# Effects of Moisture and Stone Content on the Shear Strength Characteristics of Soil-Rock Mixture

**DOI:** 10.3390/ma16020567

**Published:** 2023-01-06

**Authors:** Yu Zhang, Junyuan Lu, Wei Han, Yawen Xiong, Jinsong Qian

**Affiliations:** 1Key Laboratory of Road and Traffic Engineering of Ministry of Education, College of Transportation Engineering, Tongji University, 4800 Cao’an Road, Shanghai 201804, China; 2Key Laboratory of Infrastructure Durability and Operation Safety in Airfield of CAAC, College of Transportation Engineering, Tongji University, Shanghai 201804, China; 3Guangxi Communications Investment Technology Co., Ltd., No. 369, Minzu Avenue, Nanning 530029, China

**Keywords:** soil-rock mixture, large-scale direct shear test, shear strength, stone content

## Abstract

Soil-rock mixture is a commonly used geotechnical material used in many construction projects, such as slopes, tunnels, and dams. The shear strength of soil-rock mixture is its key property and is affected by many factors. This study aimed to investigate the shear strength characteristics of soil-rock mixture and the influences of moisture and stone content on shear strength parameters. Soil-rock mixture samples with four different stone and moisture contents were fabricated and tested using a large-scale direct shear test apparatus under four vertical pressures. The results demonstrated that the shear properties of the soil-rock mixture showed significant Mohr Coulomb failure criteria for all stone contents. As the moisture content increased, the shear strength of the soil-rock mixture first increased by 10~18% and then decreased after w = 12% to the residue value. The change in cohesion and internal friction angle of soil-rock mixture with different moisture contents shared a similar trend. For w < 12%, the cohesion and internal friction angle increased with moisture content, and for w > 12%, the two indexes obviously decreased. As the stone content increased from 30% to 60%, the shear strength of the soil-rock mixture increased by 82~174%. The internal friction angle increased linearly with stone content, while the cohesion of the mixture first increased and then decreased after the stone content reached 50%. The results can help in the designation and application of soil-rock mixture.

## 1. Introduction

Soil-rock mixture is a commonly used geotechnical material used in many construction projects, such as slopes, tunnels, and dams [1]. Soil-rock mixture is composed of “soil” and “block stone” with great differences in particle size and strength. The stone content of soil-rock mixture is generally between 25% and 75% [2]. Soil-rock mixture is composed of soil and broken rocks. Its mechanical properties are different from those of soil or broken rocks, and these properties are also not a superposition of the two materials [3].

When the content of “stone,” with large particle size and high strength, is high and there are multiple contacts to form a skeleton, the mechanical behavior of the soil-rock mixture will be controlled. Many studies have indicated that the spatial and size distribution of the stone content have a significant impact on the mechanical properties of soil-rock mixture [4,5,6]. Song et al. [7] proposed that the friction angle of soil-rock mixture increases with an increase in the particle mass fraction at particle sizes greater than 2 mm, and the greater the particle mass fraction of particles less than 0.5 mm in size, the greater the cohesion. Wu et al. [8] studied the shear mechanism and strength parameters of soil-rock mixture. Wang et al. [9] studied the particle size, surface, and shape characteristics of soil-rock mixture by digital image processing technology. Tang et al. [10] investigated the shear strength and deformation characteristics of soil-rock mixture under different freeze-thaw cycles. Nuclear magnetic resonance (NMR) was used to analyze the effect of freeze-thaw cycles on the internal pore structure of soil-rock mixture, and the relationship between pore structure and shear properties under freeze-thaw cycles was established.

Currently, the mechanical parameters and deformation features of soil-rock mixture are mainly studied by in-situ tests, laboratory tests, and numerical simulations. The mechanical parameters and deformation characteristics of soil-rock mixture are mainly controlled by its mesoscopic structure [11]. Large-scale direct shear tests have been gradually adopted to measure the strength of soil-rock mixture [12,13]. Cen et al. [14] established PFC2D random structure models with different block size distributions, calibrated the model parameters through laboratory testing, and conducted large-scale numerical shear tests at the interface. Eliadorani et al. [15] investigated the shear strength of S-RM through a series of direct shear tests at very low effective stress. A series of large-scale direct shear tests were conducted by Wei et al. [16,17].

As a typical granular material, soil-rock mixture has a mesoscale with particle size “d” as the characteristic length compared with other materials [18]. Cola et al. [19] found that the internal friction angle increased with an increase in sample size. Yin Zongze [20] proposed that the shear deformation characteristics of the sample changed with the sample size. Fu et al. [21] and Zhou et al. [22] found that a decrease in particle size caused a decrease in shear strength through laboratory experiments and numerical simulation.

This study aimed to investigate the shear strength characteristics of soil-rock mixture and the influences of moisture and stone content on shear strength parameters. Soil-rock mixture samples with four different stone and moisture contents were fabricated and tested using a large-scale direct shear test apparatus under four vertical pressures. The shear strength of soil-rock mixture under different vertical pressures, moisture contents, and stone contents were compared, and the cohesion and internal friction angle of the soil-rock mixture were calculated and analyzed.

## 2. Materials and Methods

### 2.1. Test Materials

The test material was sampled from the northwest of Guangxi Autonomous Region, China. Six representative sampling locations using soil-rock mixture were selected, as shown in Table 1.

The appearance of soil-rock mixture in natural state is shown in Figure 1, which is red-brown or yellow-brown, loose in structure, and has a large amount of gravel and irregular shape. The soil grading was tested, as shown in Figure 2.

The classification of soil in soil-rock mixture is different from the traditional concepts for sand, silt, and clay. Additionally, the grain size limit of “soil” and “stone” in soil-rock mixture depends on the research method, research conditions, etc. At present, the commonly used determination method is based on the scale independence proposed by Lindquist and Medley, as follows [23,24]:(1)$$d_{S/RT}=0.05L_{c}$$
(2)$$\left\{\begin{array}{l} d\geq d_{S/RT},stone\\ d<d_{S/RT},soil \end{array}\right.$$
where:

*S* and *RT* refers to the “soil” and “rock” in soil-rock mixture, respectively.

*L_c_* refers to the characteristic engineering scale of soil-rock mixture.

*L_c_* has different definitions for different research objects. For slope engineering, *L_c_* takes the slope height, assuming that the slope height is 20 m and *d_S/RT_* is 1 m. The soil gradation was measured using the dry screening method, the maximum particle size of the actual soil-rock mixture was far less than 1 m, and there was no boulder reaching or close to 1 m on site. It is difficult for conventional test equipment to test samples to 1 m. Obviously, this threshold is not appropriate. From the perspective of the research conditions, for direct shear test samples, *L_c_* takes the height of a single shear box of the sample. The standard size of the large shear test samples selected in research is 200 mm in diameter, 150 mm in height, the height of single shear box is 75 mm, and *d_S/RT_* is 3.75 mm, which is reasonable and basically meets other test requirements. Based on the above analysis, combined with the fractal dimension curve of the sample, the soil/stone threshold was set as 5 mm and the upper limit of particle size was 20 mm [16,17].

According to the classification standard, the natural stone content of the six samples was between 30% and 40%. Therefore, the stone contents of the laboratory test samples were determined at 30%, 40%, 50%, and 60%. The prepared test material is shown in Figure 3. The particle size analysis of soil-rock mixture was determined using the dry sieving method and the grading curve is shown in Figure 4. The particle grading and composition characteristics of the soil-rock mixture are shown in Table 2. The moisture content, density, and other test results of the soil-rock mixtures with different stone contents are shown in Table 3.

### 2.2. Test Instrument

Soil-rock mixture differs from fine-grained soil. Coarse particles constantly adjust position during the shearing process, resulting in dislocation, rolling, and shear loss. The above phenomena may play a significant role in the shear strength of soil-rock mixture. Therefore, compared with direct control of suction in the shear process of unsaturated soil -rock mixture, it is more important to prepare specimens that satisfy the size requirements and avoid the interference of size effect on the results. In this paper, a large shear tester was selected to complete the shear test. The test process met the size requirements of the soil-rock mixture specimen, and conformed to the actual stress state.

The test equipment was a YUMT-600 shear seepage coupling test machine (Tongji University, Shanghai, China), as shown in Figure 5. In the shear process, normal and shear load were applied to the specimens in the sealed box to test the shear strength under different conditions. Servo-controlled radiation flow can also be applied to the rock and soil mass to realize the research of shear seepage coupling failure under different seepage pressure. The coupling test was mostly suitable for shear failure of the rock mass structural plane. This paper mainly carried out large-scale shear tests of soil-rock mixture samples. The whole set-up of instruments and equipment was composed of the shear box, loading system, measurement, and control system.

Since the original intention of the design included the seepage problem in the sealed state, the shear box of the equipment was more complex than the general large-scale direct shear test. In addition to the shear box that directly undertook the shear function, a set of sealing boxes was also prepared, as shown in Figure 6. The lower shear box was embedded in the lower sealing box, and a copper filter screen was laid at the bottom. Water of different osmotic pressures was injected into the test piece from the center of the base through the water supply pipe, and then the test piece was discharged from the drain pipe orifice.

The loading system was composed of a dynamic and static universal testing machine, which was powered by the hydraulic press. The main machine adopted a separate portal vertical and shear loading frame, which met inside the frame and provided load application in both vertical and tangential directions at the same time. The loading was controlled by a servo system.

### 2.3. Specimen Preparation and Installation

The ratio of specimen diameter to maximum particle size was denoted as the diameter ratio (D/d_max_), and the ratio of height to maximum particle size was denoted as the height-diameter ratio (H/d_max_). The range of ratios adopted by existing research institutes include a diameter ratio of 7.5~10 and a height-diameter ratio of 4~8, while the recommended diameter ratio in the specification was 8~12 and the height diameter ratio was 4~8 [18]. The ratio of shear box (specimen size) to the maximum particle size conformed to this range. The diameter ratio was 10 and the height-diameter ratio was 7.5 in this paper.

The prepared soil-rock mixture samples with different stone contents were dried. According to the dry density, moisture content, and sample size, sufficient dry sample and water for soil-rock mixture were calculated and weighed. The dried samples were placed in a nonabsorbent basin, the amount of water added, the corresponding moisture content was calculated, and then the equipment was sprayed. After full mixing, the sample was sealed with a sealing film and wetted for 12 h. The saturated moisture content test sample was prepared in the same way. According to the specifications, sandy soil could be directly saturated in the container. The moisture content was measured (at least 2 places) at different positions of the wet sample, and the difference complied with the parallel difference allowed by the moisture content measurement. The specific steps for preparing the test piece were as following:A copper filter screen was laid at the bottom of the lower shear box. Due to the absence of osmotic pressure, water could flow out of the water inlet pipe in the center of the lower shear box during the shearing test. The prepared samples were put into the shear box in layers, and each layer was compacted to the required height with a wooden mallet.In order to stagger the shear joints, the filling height of each layer was set to 3 cm, for a total of 5 layers. For soil-rock mixture, the surface of the filled part was roughened, and then the second layer was filled. The above steps were repeated until the surface of the last layer was levelled.The shearing process of coarse-grained soil was generally accompanied by obvious dilatancy. To avoid measurement error, a certain gap was opened between the upper and lower shear boxes. The gap size recommended in the specification was (1/3~1/4) d_max_, which was set as 5 mm in this paper.After filling, the cover plate was buckled on the top surface of the mold, the upper shear box was lifted and placed into the mold, the slit ring was removed, and then the upper sealing box and its cover plate were lifted and installed. The whole set of the shear box with screws was fixed, sealed, and pushed into the predetermined test position through the lifting guide rail, as shown in Figure 7.The dynamic and static universal testing machine was started, the load threshold through the control software was set, the vertical jack was started to make each part contact, and it was set to “zero.”. Then, the horizontal jack was started to make contact with the tangential dowel rod to stop.For the samples prepared for each group of tests, the density difference was not greater than 0.03 g·cm^−3^, and the moisture content difference was not greater than 1%. The tests were carried out under different pressures, and the vertical pressure differences at all levels were roughly the same.

### 2.4. Test Procedure

Considering the high permeability of soil-rock mixture, the quick shear mode was used in the shear process in this paper. The vertical pressures were taken as 100, 200, 300, and 400 kPa respectively, and the loading rate was set at 10 kPa·s^−1^. After the vertical stress was stable, if the vertical deformation per hour was less than 0.03 mm, it was assumed that the deformation was stable. When the shear displacement under a certain level of load exceeded 1.5~2.0 times the previous level of shear displacement, it was applied according to 5% of the corresponding vertical load. The whole process was not less than 10 levels. Each level was applied for 30 s, and the loading rate was also 10 kPa·s^−1^. The actual applied load increased the frictional resistance. The ultimate strength standard was used as the failure criterion. The failure criterion was when the horizontal load was stable or the shear deformation increased sharply. If there was no obvious peak value of the horizontal load, the test was stopped when the shear displacement reached 2 cm, which was 1/10 of the sample diameter.

In addition to the stone content, the moisture content of the sample is also an important factor affecting the shear strength [25]. For the shear strength of soil-rock mixture, both soil and stone play a role, resulting in the decline of applicability. Therefore, samples with different stone contents were selected to form the specimens, and the moisture contents (w) were set at 6%, 12%, 18%, and 24% (saturated). Considering the different vertical pressure levels for the shear test, 64 groups of shear tests were conducted in this paper, as shown in Table 4.

## 3. Results and Discussion

### 3.1. Shear Failure Characteristics of Soil-Rock Mixture

Representative test data were selected and the shear stress shear displacement relationship curve was drawn, as shown in Figure 8, to show the impact of different stone contents, vertical pressures, and moisture contents on the shear failure process.

As shown in Figure 8, the critical state of the soil-rock mixture was not uniform. In most cases, it showed strain softening. When the stone content was low and the vertical pressure was small, it showed strain hardening. However, the above two critical states did not affect the determination of shear strength. More importantly, unlike the smooth stress displacement curve of fine-grained soil, the phenomenon of “stress fluctuation” occurred in the shear process of soil-rock mixture. This phenomenon can be described as the sudden drop of shear stress in the shear process, which then quickly recovers or exceeds the level before the drop. This phenomenon was reflected in varying degrees in each stage of each group of shear tests. When the stone content was high (C_*r*_ > 50%), the vertical stress was high (P_*v*_ > 300 kPa), and the moisture content was low (w < 12%, S < 0.5), the stress fluctuation was particularly significant.

The irregular shape of the shear surface of soil-rock mixture is caused by the large particle size difference between “stone” and “soil”. Under the combined action of vertical pressure and shear stress, the originally loose skeleton structure begins to make contact and become embedded. With the continuous shear process, the coarse particles forming the skeleton become staggered or broken, and the continuous reconstruction of the skeleton structure and filler causes the undulation of the shear surface, which is consistent with the stress fluctuation phenomenon.

### 3.2. Effects of Vertical Pressure on Shear Strength

The shear failure surface was fitted to calculate the ordinate intercept and slope angle, which are represented by cohesion and the internal friction angle, respectively. The shear strength can be calculated using Equation (3):(3)τ=σtanφ+c
where:

*τ* is the shear strength;

*σ* is the normal stress, corresponding to the vertical pressure in the shear test;

*c* is cohesion;

*φ* is the internal friction angle.

The test results are shown in Figure 9. The shear strength of soil-rock mixture increased significantly with the vertical shear pressure and moisture content. The shear strength parameters were calculated by fitting the Mohr Coulomb strength failure envelope. When the vertical pressure increased from 100 to 400 kPa, the shear strength increased by approximately 189~229%. Under different moisture content conditions, the response of shear strength to vertical pressure change was relatively stable. The increase in shear strength in the high moisture content specimen was slightly higher, and the increase was linearly related to the increasing degree of moisture content. The response of the vertical pressure of the shear strength changed abruptly under different stone content conditions. When the stone content was in the range of 30~50%, the increase in shear strength was relatively consistent. When the stone content was 60%, the increase in shear strength increased significantly, which confirmed that the structural characteristics of soil-rock mixture changed when the stone content approached 60%.

### 3.3. Effects of Moisture Content on Shear Strength

The test results are shown in Figure 10. It can be seen that with the increase in moisture content, the change rule of shear strength of soil-rock mixture was similar under any stone content and vertical pressure. Unlike the common monotonic decline of fine-grained soil, the shear strength of soil-rock mixture first increased and then decreased in the set range. When the moisture content was in the range of 6~12%, the shear strength increased and reached the peak at 12%. When the moisture content was in the range of 12% to saturation, the shear strength decreased to the stable strength, and the decline process was first fast and then slow.

This phenomenon stems from the complex influence of water on the strength of geotechnical materials, and the special structural characteristics of soil-rock mixture further aggravate the complexity. Based on the mechanics of unsaturated soil, the greatest characteristic of unsaturated soil is the existence of matrix suction. Matrix suction may give an additional strength that can be interpreted as apparent cohesion, but not additional friction. Indeed, it causes an increase in effective stress and consequently strength. Based on the water film theory, there is true cohesion in viscous particles, which differs from apparent cohesion. This is the result of a series of physical and chemical actions, which are related to the structure and composition of clay minerals and the thickness of adsorbed water film between particles [26].

The structure of soil-rock mixture is relatively loose, and its soil water characteristic curve differs from that of silt or clay. The air inlet value is small and the transition section is steep [27]. The increase in matrix suction was generally small in the saturation range corresponding to a moisture content range of 6~12%. According to the water film theory, the water and air in soil-rock mixture in the transition section are connected and can flow. They belong to a double open system. Surface tension is formed as the film shrinks at the air-water interface, and its reaction force acts on the soil-rock particles and produces compressive stress on the particles, resulting in an increase in soil cohesion [28]. The results showed that the water film action dominated the shear strength of the soil-rock mixture in this range, which increased slightly to the peak strength. As the moisture content continued to increase, the matrix suction of the soil-rock mixture continued to decrease until it dropped to 0. The film shrinkage at the air-water interface was also reduced accordingly. From the perspective of true cohesion, the wedging effect of water on the cementitious material in the soil leads to its destruction. Subsequently, the water film begins to show a lubrication effect, decreasing the true cohesion between the soil and rock particles, and facilitating the sliding of particles. The results showed that the joint action of the two effects led to the rapid decline of the shear strength of the soil-rock mixture to the stable strength.

When the moisture content increased from 6% to 12%, the shear strength increased by approximately 10~18%. When the moisture content continued to increase to saturation, the shear strength decreased to 48~73% of the shear strength under the 6% moisture content condition.

### 3.4. Effects of Stone Content on Shear Strength

The test results are shown in Figure 11. It can be seen that with the increase in stone content, the change in shear strength of soil-rock mixture was similar under all moisture contents and vertical pressure. With the increase in stone content, the shear strength of soil-rock mixture increased significantly, and the four measuring points of specific moisture content and vertical pressure showed a good linear relationship. When the stone content increased from 30% to 60%, the shear strength increased by approximately 82~174%. Under different moisture content conditions, the response of shear strength to the change in stone content showed different characteristics. The increase in the shear strength of the soil-rock mixture with high moisture content was significantly higher than that of the soil-rock mixture with low moisture content. In the saturated state, the maximum increase in shear strength was approximately 170%, while under the 6% moisture content condition, the maximum increase in shear strength was approximately 80%. In the soil-rock mixture, the shear strength of the fine particles is greatly affected by water, while the coarse particles can form a skeleton structure, and the shear strength dominated by this structural feature is less affected by the action of water [29]. Therefore, under the condition of high moisture content, the shear strength of the sample with low stone content was significantly lower, while the shear strength of the sample with high stone content was locked between coarse particles, and the decrease in shear strength was small. The combination of the two trends showed that the higher moisture content resulted in increased shear strength with increasing stone content.

Under different stone content conditions, the response of shear strength to the change in moisture content was not consistent. In the rising section of shear strength, the higher stone content resulted in a greater increase in peak shear strength. The results showed that the higher stone content resulted in a steeper transition section corresponding to the soil water characteristic curve, and the decrease in matrix suction caused by the increase in moisture content was small. At the same time, it was speculated that the surface tension of the high stone content test material under the action of the water film was stronger at this stage, and the combined action of the two effects further promoted the peak strength of the high stone content test material. In a matrix-sustained field, a gradual decrease in friction was observed with an increase in the fine particle content [30]. In the decreasing section of shear strength, the shear strength of the low stone content soil-rock mixture decreased to a greater extent and more sharply. The results showed that the control effect of the fine particles in the low stone content sample was more significant, and it was more in line with the conclusion that the joint action of apparent cohesion and true cohesion lead to the decrease in shear strength. For the soil-rock mixture with high stone content, the coarse particles played a controlling role in the residual stage, and the embedded and locking effects at the macro structure level eased the decline range and rate of shear strength to a certain extent.

### 3.5. Effects of Moisture Content on the Shear Strength Parameters

The relationship between moisture content and shear strength parameters is shown in Figure 12. It can be seen that under different stone content conditions, the change trend of cohesion with the increase in moisture content was relatively close, showing a trend of first rising and then falling. The cohesion reached the peak near the range of 10~12% moisture content. In the rising stage of cohesion, the increase was more significant for the high stone content specimen, while in the falling stage of cohesion, the decrease was more significant for the low stone content specimen.

It can be seen from Figure 12b that under different stone content conditions, the change trend of internal friction angle with the increase in moisture content was highly consistent, showing the same trend of first rising and then falling, and its peak value was similar to the peak range of cohesion, in the range of 10~12% moisture content. When the moisture content of the sample was lower than the optimal moisture content, the increase in moisture can make the relatively loose structure of the soil-rock mixture more compact and improve the internal friction angle. When the moisture content continued to increase, water played a lubricating role and softened between particles, resulting in a decrease in internal friction angle. When calculating the change range, it was found that the maximum decrease range of the internal friction angle from low stone content to high stone content was 49%, 38%, 29%, and 24%, indicating that the change in moisture content had a more significant impact on the soil-rock mixture with low stone content. The results showed that there were more fine particles in the test material with low stone content. When the moisture content increased, the lubrication and softening effect were strengthened, and the bite form between particles was also affected. As a result, the internal friction angle decreased significantly while there was a skeleton structure in the high stone content test sample. The increase in water did not fundamentally change the bite form between particles, so the decrease in the internal friction angle was significantly lower.

### 3.6. Effects of Stone Content on the Shear Strength Parameters

The relationship curve between stone content and shear strength parameters is shown in Figure 13. It can be seen from Figure 13a that under the conditions of different moisture content, the change trend of cohesion with the increase in stone content was similar, showing the trend of first increasing and then decreasing, and reaching the peak at about 50% of stone content. The results showed that this change trend was consistent with the structural change caused by the increase in stone content in soil-rock mixture. With the increase in stone content, the fine particles decreased, the suction dominated by the fine particles decreased, and the bite dominated by coarse particles increased. When the stone content was less than 50%, the soil-rock mixture presented a dense skeleton structure. Even if the proportion of fine particles was reduced, it remained dense, which provided a certain matrix suction and acted together with the coarse particles to maintain the increase in cohesion. When the stone content exceeded 50%, the soil-rock mixture transitioned to the skeleton void structure, and the matrix suction provided by the fine particles decreased significantly, exceeding the increase in the bite effect caused by the increase in coarse particles, and finally resulting in a decrease in cohesion. It can also be concluded from the variation range of samples with different moisture contents that the matrix suction of the fine particles in saturated or high moisture content samples was small, and the cohesion was mainly provided by the bite of coarse particles. Thus, the “peak” phenomenon was not significant. On the contrary, the phenomenon of “peak value” was significant in the low moisture content samples.

It can be seen from Figure 13b that under different moisture content conditions, the variation trend of internal friction angle with the increase in stone content was highly consistent, showing a clear positive correlation. The results showed that due to the increase in stone content, the skeleton of the soil-rock mixture was gradually formed, and the intercalation and occlusion between particles were improved [27]. Therefore, the internal friction angle increased linearly. The maximum increase of different samples with moisture contents ranging from 6% to saturation increased by 80.1%, 85.4%, 141.3%, and 168.7%, respectively. The higher the moisture content, the greater the increase. The results showed that the influence of water on the bite state between fine particles was higher than that of coarse particles. Under the condition of high moisture content, the bite effect between fine particles was almost lost, and the friction angle was basically provided by coarse particles. With the increase in coarse particles, the increase in the friction angle of the soil-rock mixture was more significant.

## 4. Conclusions

This paper conducted a series of laboratory tests to investigate the shear strength characteristics of soil-rock mixture and the factors influencing the shear strength parameters. Based on the test results, the following conclusions can be drawn:

Different from the smooth stress displacement curve of fine-grained soil, the phenomenon of “stress fluctuation” occurs in the shear process of soil-rock mixture. When the stone content is high (C_*r*_ > 50%), the vertical stress is high (P_*v*_ > 300 kPa), and the moisture content is low (w < 12%, S < 0.5), the stress fluctuation is particularly significant.

The shear strength of soil-rock mixture increases linearly with the increase in vertical pressure. As the vertical pressure increased from 100 to 400 kPa, the shear strength increased by approximately 189~229%.

As the moisture content increased, the shear strength of the soil-rock mixture first increased by 10~18% and then decreased after w = 12% to the residue value. The mixture with higher stone content tended to be more sensitive to the change in moisture content.

The change in cohesion and internal friction angle of the soil-rock mixture samples with different moisture contents shared a similar trend. For w < 12%, the cohesion and internal friction angle increased with moisture content, and for w > 12%, the two indexes obviously decreased.

As the stone content increased from 30% to 60%, the shear strength of the soil-rock mixture increased by approximately 82~174%. The internal friction angle increased linearly with the stone content, while the cohesion of the soil-rock mixture first increased and then decreased after the stone content reached 50%.

## Figures and Tables

**Figure 1 materials-16-00567-f001:**
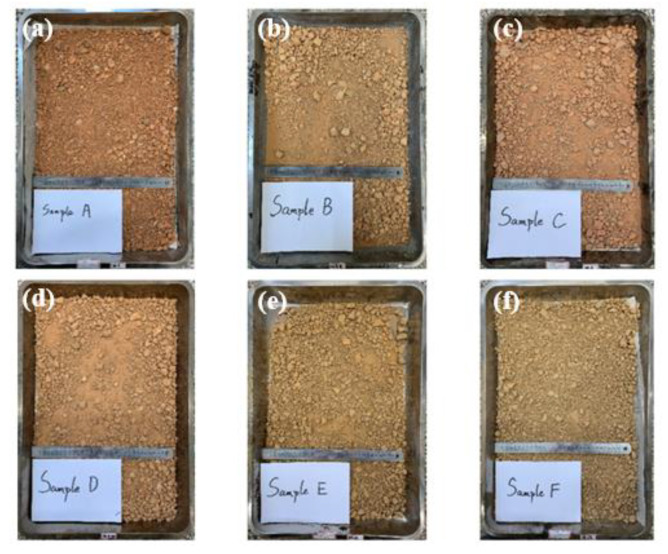
Samples of soil-rock mixture from six different locations.

**Figure 2 materials-16-00567-f002:**
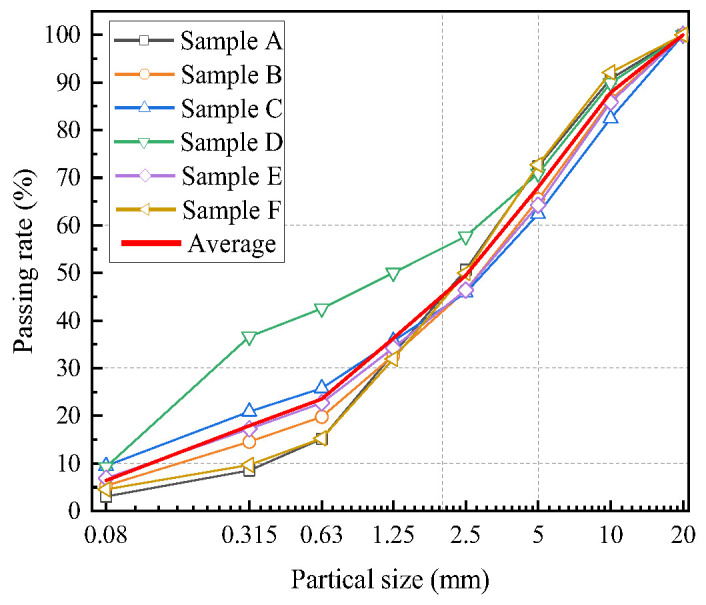
Grading curves of the six soil-rock mixture samples.

**Figure 3 materials-16-00567-f003:**
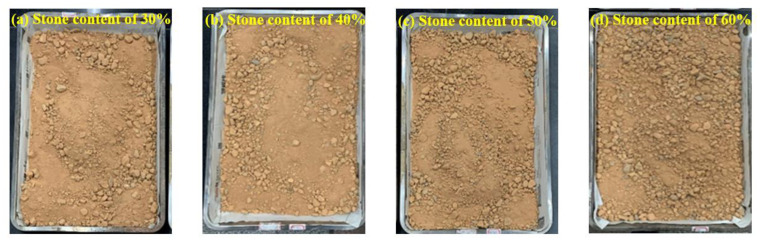
Samples with different stone contents.

**Figure 4 materials-16-00567-f004:**
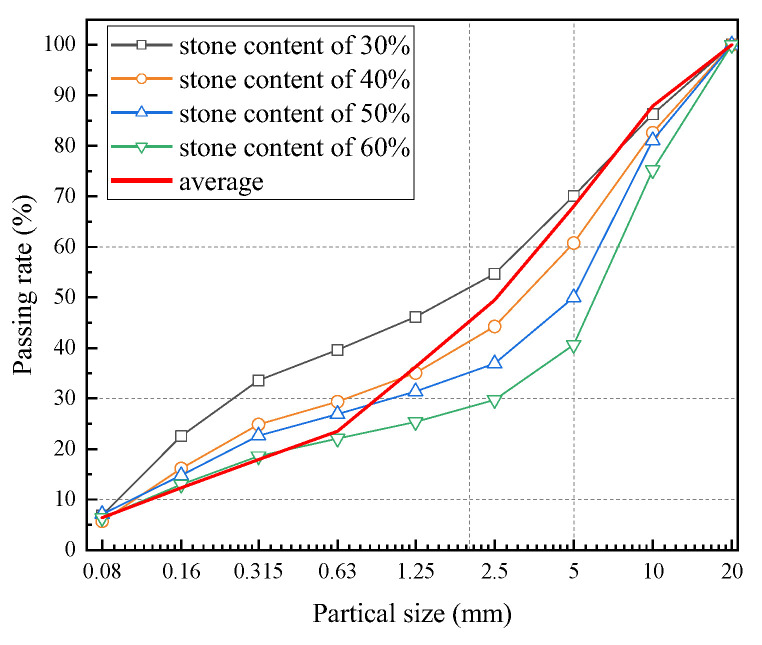
Grading curve of samples with different stone contents.

**Figure 5 materials-16-00567-f005:**
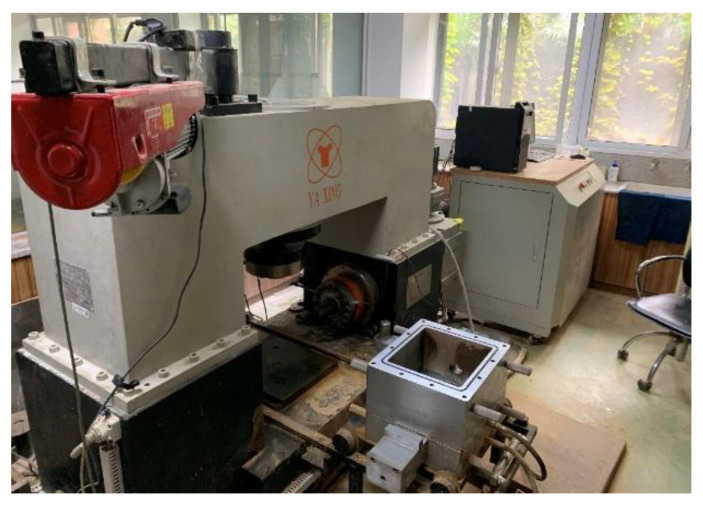
Shear seepage coupling testing machine.

**Figure 6 materials-16-00567-f006:**
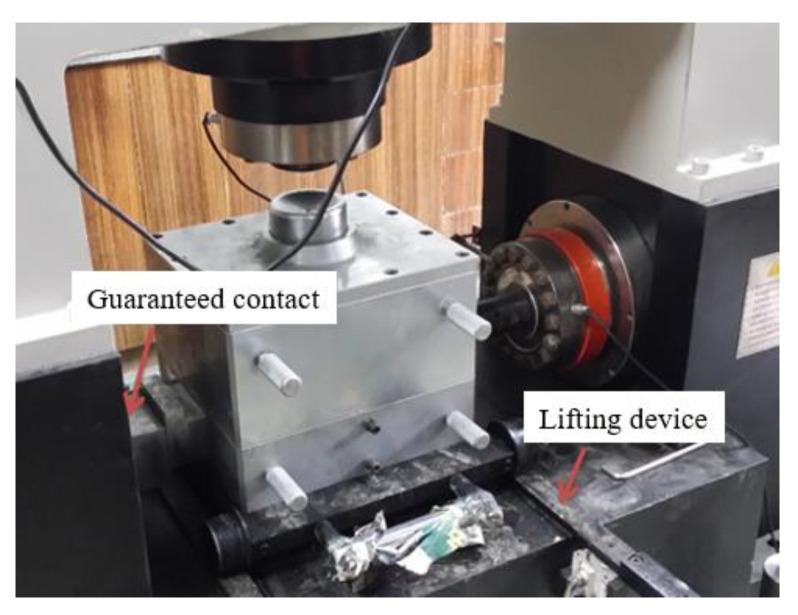
Large shear test shear box.

**Figure 7 materials-16-00567-f007:**
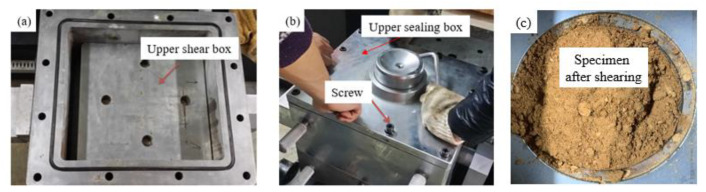
Installation process of complete shear box. (**a**) Upper shear box (**b**) Upper sealing box and screw (**c**) Specimen after shearing.

**Figure 8 materials-16-00567-f008:**
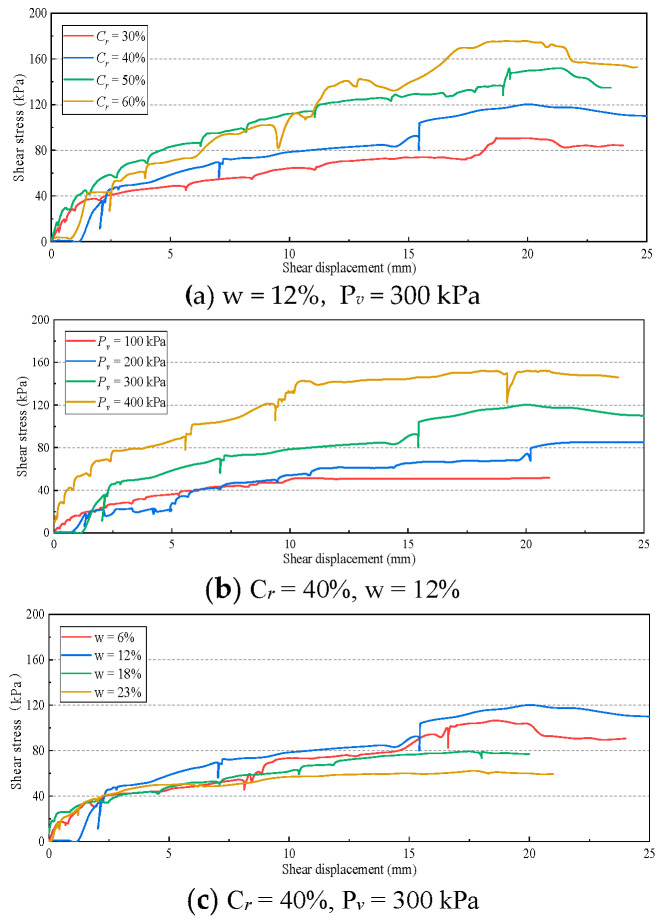
Shear stress shear displacement curve of soil-rock mixture. (**a**) w = 12%, P_*v*_ = 300 kPa (**b**) C_*r*_ = 40%, w = 12% (**c**) C_*r*_ = 40%, P_*v*_ = 300 kPa.

**Figure 9 materials-16-00567-f009:**
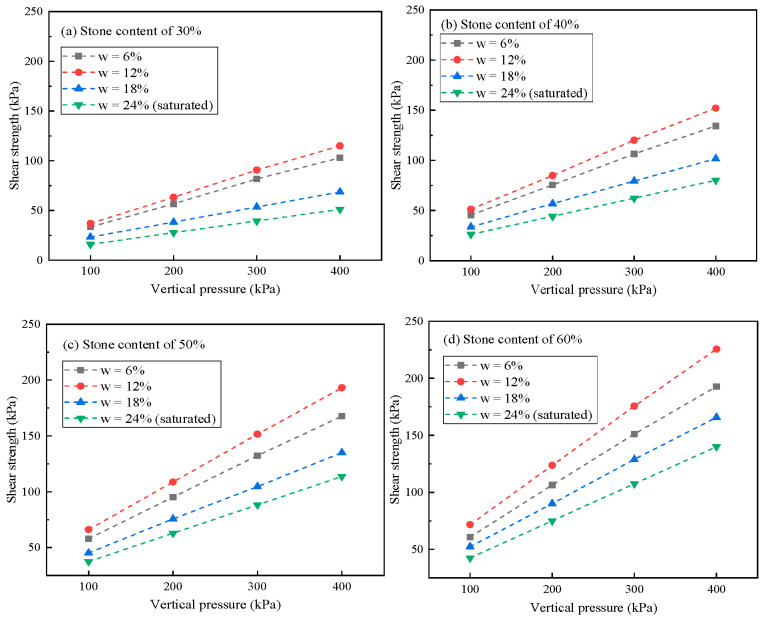
Shear strength under different vertical pressures.

**Figure 10 materials-16-00567-f010:**
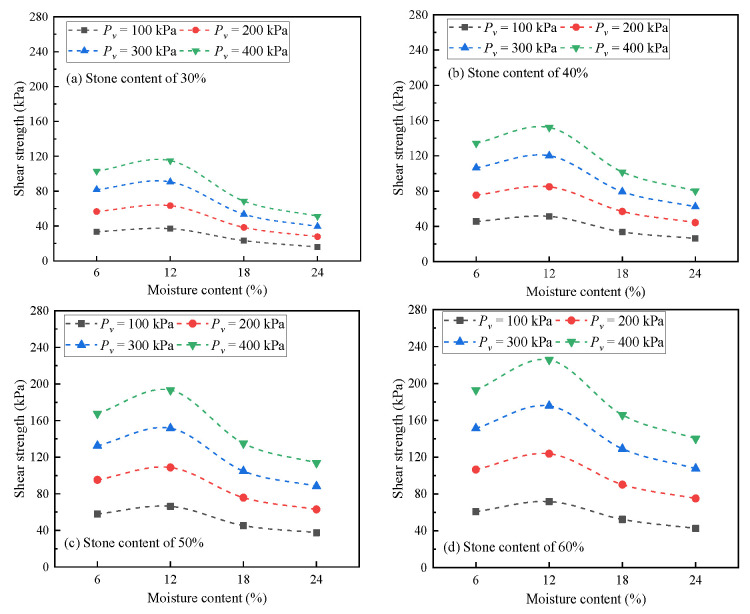
Shear strength under different moisture content.

**Figure 11 materials-16-00567-f011:**
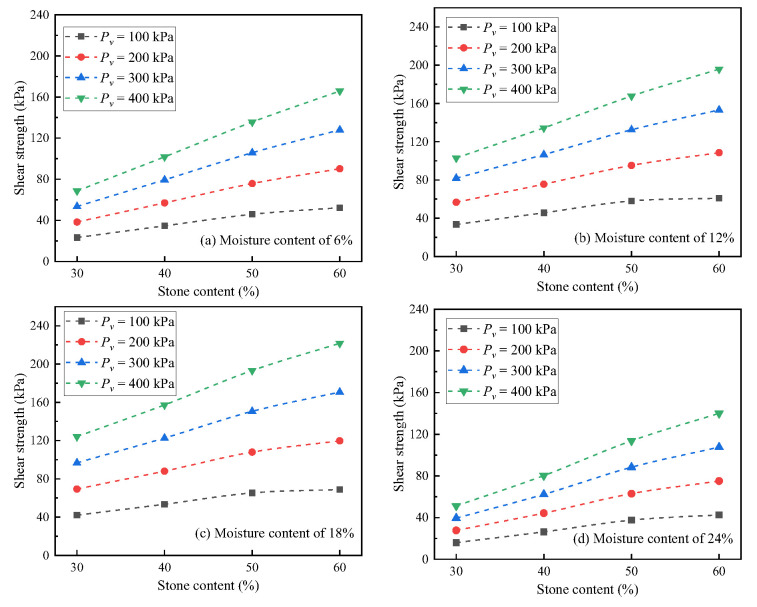
Shear strength under different stone contents.

**Figure 12 materials-16-00567-f012:**
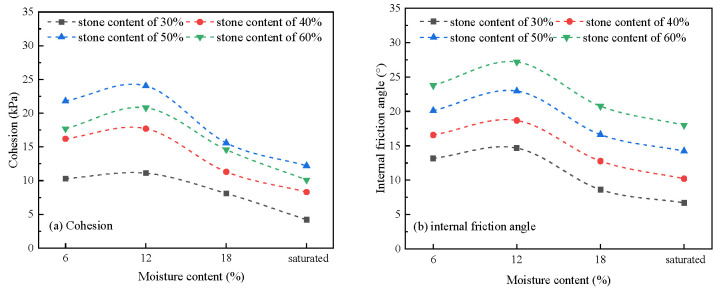
Shear strength parameters under different moisture content.

**Figure 13 materials-16-00567-f013:**
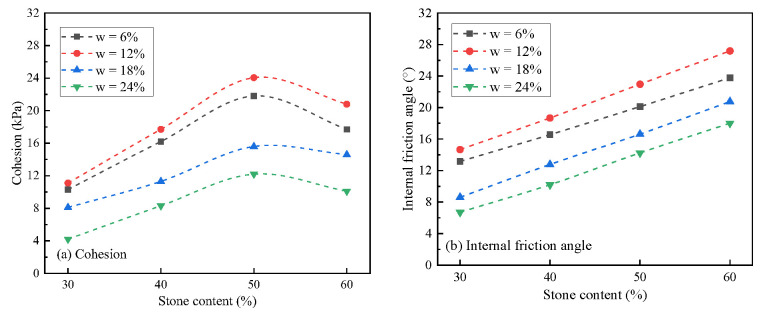
Shear strength parameters under different stone contents.

**Table 1 materials-16-00567-t001:** Sampling locations of soil-rock mixture of supporting projects.

Index	Sampling Location	Mileage Stake No.
A	Lingzhan Toll Station	JK0 + 200
B	Yongle Tunnel	K174 + 310~600
		K182 + 150
C	Naxian Tunnel	K144 + 540
D	Lingyun Toll Station	K93 + 800
E	Tianwan Tunnel	YK54 + 330~YK56 + 792
F	Leye Toll Station	K64 + 660

**Table 2 materials-16-00567-t002:** Grain gradation of test materials with different stone contents.

Stone Content (%)	D_60_	D_30_	D_10_	C_U_	C_C_
30%	2.93	0.23	0.09	32.56	0.19
40%	3.50	0.25	0.09	39.81	0.20
50%	6.12	0.63	0.10	61.79	0.66
60%	6.89	0.90	0.10	71.07	1.22
Average	3.74	0.90	0.19	28.14	1.33

**Table 3 materials-16-00567-t003:** Density and moisture content of samples with different stone contents.

Stone Content (%)	Natural Bulk Dry Density (g·cm^−3^)	Saturation Density (g·cm^−3^)	Saturated Moisture Content (g·cm^−3^)
30%	1.55	1.92	24
40%	1.60	1.96	23
50%	1.63	2.03	25
60%	1.67	2.14	28

**Table 4 materials-16-00567-t004:** List of shear test variables for soil rock mixture.

Vertical pressure P*_v_* (kPa)	100; 200; 300; 400
Stone Content C*_r_* (%)	30; 40; 50; 60
Moisture Content w (%)	6; 12; 18; 24

## Data Availability

Not applicable.

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
