# Peer review of "Effects of Moisture and Stone Content on the Shear Strength Characteristics of Soil-Rock Mixture"

_materials, 2023, doi:10.3390/ma16020567_

Round 1
Reviewer 1 Report
The paper aims to investigate the shear strength characteristics of the soil-rock mixture, evaluating the influence of moisture content and stone content on the shear strength parameters.
For this purpose, samples of soil-rock mixture with four different stone contents and four moisture contents were manufactured and tested under direct shear.
Indicate in the abstract the contribution of the study (mandatory item).
The introduction contains much repetition.
The text needs to make clear the knowledge gap. Improve.
It can be expanded, including new references.
Method
Details on the execution of the tests need to be included.
How was the granulometry of the particles performed (fig 1)?
Essential information for the characterization of soils and mixtures needs to be included.
What is the compression index? What is the demand for water? What is the liquidity limit? What is the plasticity limit?
In the results and discussions, improve the correlation with the literature.
Keep the graphics in figures 6, 7, 8, and 9 on the same scale - this way, it can compare the samples more efficiently.
The theoretical framework is weak. It could include new studies.
Some minor comments were made in the body of the text.

Reviewer 2 Report
The paper is very descriptive, merely a description of results, but the analysis is very poor and the interpretation of thr findings is doubtful.
The behaviour of mixtures has been recently analysed by a number of papers in the geotechnical literature, among the most relevant:
- Thevanayagam, S. & Mohan, S. (2000). Intergranular state variables and stress–strain behaviour of silty sands. Géotechnique 50,No. 1, 1–23, http://dx.doi.org/10.1680/geot.2000.50.1.1.
- Wood, D. M. & Kumar, G. V. (2000). Experimental observations of behaviour of heterogeneous soils. Mech. Cohesive–Frictional Mater. 5, No. 5, 373–398.
- P. Ruggeri, et al Evaluating the shear strength of a natural heterogeneous soil using reconstituted mixtures. Géotechnique, 2016. http://dx.doi.org/10.1680/jgeot.15.P.022]
1) It is shown that to interpretate results from testing heterogeneous soils critical parameter such as the fine content and the specific volume of the granular fraction are needed. This is because the behaviour of the mixture depends on the available volume that can be occupied by the fine and how much fine is included in this volume.
- In the paper the above parameters are not all considered. Instead, there is an extensive use of the water content and of the stone content;
- It is suggested to represent results so that the change of the fine content is considered for a fixed specific volume of the granular fraction
- Non saturation is strongly complicating the interpretation of results; saturation degree for any of the tests shall be given
- Introduce a Table with all information about the samples, such as saturation degree, water content, fine content, and information from testing such as specific volume at the end of consolidation and any useful information to evaluate the physical state of samples
2) The testing arrangement and the procedure are not clearly described:
- What is dS/RT ? Give definition of “d”
- Why define a “diameter ratio “if the sample is prismatic ?
- Is the test strain or stress controlled ?
- Show the stress path to failure if known and the corresponding stress-displacement plot
Are stresses effective stresses ? but with non-saturation are the effective stresses really known ? (Conclusion n. 1 can be claimed only if effective stresses are known
Round 2
Reviewer 1 Report
The authors made the suggested corrections, which improved the quality of the paper.
This could be published.
Reviewer 2 Report
see the attached file

Round 3
